# On fuzzy Henstock-Stieltjes integral on time scales with respect to bounded variation function

Juan Li[1], Yubing Li[2], Yabin Shao 🔟 [2]*

**1** School of Mathematics and Information Science, Baoji University of Arts and Sciences, Baoji, Shannxi, China, **2** School of Science, Chongqing University of Posts and Telecommunications, Chongqing, China

* shaoyb@cqupt.edu.cn

**Data Availability Statement:** All relevant data are within the manuscript.

**Funding:** The author(s) received no specific funding for this work.

## Abstract

In present paper we will investigate the basic theory of fuzzy Henstock-Stieltjes Δ-integral with respect to a bounded variation function on time scale. Firstly, we define the notion of fuzzy Henstock-Stieltjes Δ-integral (or briefly FHS-Δ-integral) on time scales, and propose some basic properties and several necessary and sufficient conditions for fuzzy Henstock-Stieltjes Δ-integrable functions. Secondly, we present a characterization theorem of fuzzy Henstock-Stieltjes Δ-integrable function by using the embedding theorem of fuzzy number space. Therefore, this paper complements and enriches the theory of fuzzy integral, and the results of this paper will contribute to establishing discontinuous fuzzy dynamic equations on time scales.

## Introduction

Fuzzy phenomena are common in real world. It is difficult for people to deal with fuzzy phenomena with traditional methods and existing tools. Since 1965, Zadeh [1] first introduced the notion of fuzzy sets, which has been studied widely from the theories and applications, such as fuzzy topology, fuzzy analysis, fuzzy decision making, robotics, medicine, biological sciences, operations research, image processing, and so on. In recent years, many authors studied fuzzy integrals from different points of view, their research not only makes the fuzzy integral theory more perfect, but also deals with the relevant problems of fuzzy differential equations [2, 3]. Such as, due to the usual notion of differentiability of fuzzy functions has a defect: if $\tilde{m} \in \mathbb{R}_{\mathcal{F}}$ and $g : [a, b] \to \mathbb{R}$ is differentiable with $g'(x) \leq 0$, then $\tilde{f}(x) = \tilde{m} \cdot g(x)$ is not differentiable. To solve this shortcoming, Bede and Gal [4] proposed the generalizations of the differentiability of fuzzy-number-valued functions. To study convex fuzzy programming, Wang et al. [5] discuss the notions of directional derivative, differential and sub-differential for fuzzy-mapping and given the relationship between them in 2023.

As we all know, Nanda [6] proposed the notion of the Riemann-Stieltjes integral for fuzzy-number-valued functions in 1989. Since then, Wu et al. [7] found that bounded fuzzy number sets have supremum and infimum, but the supremum and infimum can't be approximated arbitrarily, and it is not necessarily reachable for continuous functions. That is, the concept of

**Competing interests:** The authors have declared that no competing interests exist.

fuzzy Riemann-Stieltjes integral proposed by Nanda was incorrect. Therefore, this fact has stimulated people's interest to overcome this limitation. In 1998, Wu [8] introduced the notion of the fuzzy Riemann-Stieltjes integral, which found that we can solve a nonlinear programming problem and obtain its membership function, but it is not easy to generalize and calculate to a high-dimensional space. In 2006, Ren et al. [9] proposed a new kind of fuzzy Riemann-Stieltjes integral and they discussed properties of the integral. This integral is defined for the continuous fuzzy-number-valued function. However, we find that if a fuzzy-number-valued function is discontinuous or not fuzzy Riemann-Stieltjes integrable, we can't deal with it directly with the existing methods.

To further develop the integral theory of real-valued function, Denjoy and Perron proposed Denjoy integral [10] and Perron integral [11]. Though Denjoy integral and Perron integral generalized Lebesgue integral, they had many shortcomings such as it was not easy to handle some problems using their definitions. Until 1957-1963, the Riemann-type definition was introduced by Kurzweil and Henstock, respectively; the current definition is simpler than previously proposed. As we all know, the Henstock integral [12] includes the Riemann, improper Riemann, Lebesgue and Newton integrals. In 1998, Lim et al. [13] defined Henstock-Stieltjes integral of real-valued functions with respect to an increasing function. In 2000, Wu and Gong [14, 15] studied the fuzzy Henstock integral. In 2012, Gong and Wang [16] introduced Henstock-Stieltjes integral for fuzzy-number-valued functions. Later, we [17–19] researched this integral deeply and formed a relatively complete integral theory.

In order to unify and generalize continuous and discrete dynamical systems, Hilger [20] first proposed the concept of time scales in his Ph.D. thesis in 1988. It is worth mentioning that Vipin Kumar et al. [21] studied the existence of solution, stability analysis and exact controllability results for an abstract integro hybrid evolution system with impulses on time scales. There are many applications which have some jumps at some specific time moments. So, Vipin Kumar et al. [22] considered an impulsive switched system and established the finite time stability (FTS) results. In [23], the authors investigated the projective lag quasi-synchronization by feedback control of a coupled dynamical system with delays and parameter mismatches on arbitrary time domains. In 2023, they gave a more general conclusion, extending the above theorems to on arbitrary time domain [24]. Around 2000, the calculus theory of time scale was established [25, 26]. After that, Peterson and Thompson [27] introduced the Henstock delta integral on time scales. In 2009, Mozyrska et al. [28] defined and gave some basic properties for the Riemann-Stieltjes integral on time scales, which makes it possible to further study Fredholm and Volterra-Stieltjes integral equations on time scales. In recent years, people have studied calculus on time scales deeply [29, 30]. In 2016, Yoon [31] introduced the Henstock-Stieltjes integrals of interval-valued functions on time scales. In 2019, Zhao et al. [32] introduced the interval Darboux delta integral and the interval Riemann delta integral on time scales. As the generalization theory of the interval Riemann delta integral, Muawya [33] introduced the notion of the McShane-Stieltjes integrals of interval-valued functions and fuzzy-number-valued functions on time scales. In 2021, Afariogun et al. [34] proposed the Henstock-Kurzweil Stieltjes double integrals of interval-valued functions on time scales and proved some of the basic properties of this integral. However, the above results are all one-sided research from one aspect. Therefore, in order to further enrich the integral theory, this paper will systematically study the fuzzy Henstock-Stieltjes Δ-integral on time scales. The main contributions of this paper are as follows:

(1) Some essential properties of fuzzy Henstock-Stieltjes Δ-integral on time scales are established.

(2) Through an isometric isomorphism embedding operator, a characterization theorem of FHS-$\Delta$-integrable functions is obtained.

The paper is organized as follows. The "Preliminaries" section provide some basic notions and preliminary results of fuzzy numbers and time scales. "The fuzzy Henstock-Stieltjes $\Delta$-integral on time scales" section, we define the fuzzy Henstock-Stieltjes $\Delta$-integral on time scales. At the same time, we obtain some properties and a necessary and sufficient conditions of the fuzzy Henstock-Stieltjes $\Delta$-integrable function on time scales. "The characterization of FHS-$\Delta$-integrable functions" section, we obtain a characterization theorem for fuzzy Henstock-Stieltjes $\Delta$-integrable functions by the Vector-Henstock-Stieltjes $\Delta$-integral on Banach space. Finally, the "Conclusions" section, we give some conclusions and future research.

## Preliminaries

In this section, some preliminary notion of fuzzy numbers and time scales are introduced as a prework, including operations and metric of fuzzy numbers and properties of time scales.

Let us denote by $\mathbb{R}_{\mathcal{F}}$ the class of fuzzy subsets of the real axis (i.e. $u : \mathbb{R} \to [0, 1]$) satisfying the following properties:

(1) $u$ is normal, i.e. there exists $x_0 \in \mathbb{R}$ with $u(x_0) = 1$;

(2) $u$ is convex fuzzy set (i.e $u(tx + (1 - t)y) \geq \min\{u(x), u(y)\}, \forall t \in [0, 1], x, y \in \mathbb{R}$);

(3) $u$ is upper semicontinuous on $\mathbb{R}$;

(4) $\overline{\{x \in \mathbb{R} : u(x) > 0\}}$ is compact, where $\overline{A}$ denotes the closure of $A$.

Then $\mathbb{R}_{\mathcal{F}}$ is called the space of fuzzy numbers. For $0 < r \leq 1$, denote $[\tilde{u}]_r = \{x \in \mathbb{R} : u(x) \geq r\}$ and $[u]_0 = \{x \in \mathbb{R} : u(x) > 0\}$. As we all know, for any $r \in [0, 1], [\tilde{u}]_r$ is a bounded closed interval. For $\tilde{u}, \tilde{v} \in \mathbb{R}_{\mathcal{F}}$ and $k \in \mathbb{R}$, the sum $\tilde{u} + \tilde{v}$ and the product $k \cdot \tilde{u}$ are defined by $[\tilde{u} + \tilde{v}]_r = [\tilde{u}]_r + [\tilde{v}]_r, [k \cdot \tilde{u}]_r = k \cdot [\tilde{u}]_r, \forall r \in [0, 1]$, where $[\tilde{u}]_r + [\tilde{v}]_r = \{x + y : x \in [\tilde{u}]_r, y \in [\tilde{v}]_r\}$ means the usual addition of two intervals and $k \cdot [\tilde{u}]_r = \{kx : x \in [\tilde{u}]_r\}$ means the usual product between a scalar and a subset of $\mathbb{R}$ (see e.g. [19]).

Define $D : \mathbb{R}_{\mathcal{F}} \times \mathbb{R}_{\mathcal{F}} \to \mathbb{R}_+ \cup \{0\}$ by

$$D(\tilde{u}, \tilde{v}) = \sup_{r \in [0,1]} \max\{| u_r^- - v_r^- |, | u_r^+ - v_r^+ |\},$$

where $[\tilde{u}]_r = [u_r^-, u_r^+], [\tilde{v}]_r = [v_r^-, v_r^+]$. $D(\tilde{u}, \tilde{v})$ is called the distance between fuzzy numbers $\tilde{u}$ and $\tilde{v}$. Using the results in [29, 35], we know that

(1)$(\mathbb{R}_{\mathcal{F}}, D)$ is a complete metric space;

(2)$D(\tilde{u} + \tilde{w}, \tilde{v} + \tilde{w}) = D(\tilde{u}, \tilde{v})$;

(3)$D(k \cdot \tilde{u}, k \cdot \tilde{v}) = | k | D(\tilde{u}, \tilde{v}), k \in \mathbb{R}$;

(4)$D(\tilde{u} \cdot \tilde{v}, \tilde{u} \cdot \tilde{w}) \leq D(\tilde{u}, \tilde{0})D(\tilde{v}, \tilde{w})$;

(5)$D(\tilde{u} + \tilde{v}, \tilde{w} + \tilde{e}) \leq D(\tilde{u}, \tilde{w}) + D(\tilde{v}, \tilde{e})$, where $\tilde{u}, \tilde{v}, \tilde{w}, \tilde{e} \in \mathbb{R}_{\mathcal{F}}$.

In addition, we introduce a partial order in $\mathbb{R}_{\mathcal{F}}$ by $\tilde{u} \leq \tilde{v}$ iff $[\tilde{u}]_r \leq [\tilde{v}]_r, r \in [0, 1]$ iff $u_r^+ \leq v_r^+, u_r^- \leq v_r^-, r \in [0, 1]$ (see e.g. [36]).

The representation theorem of fuzzy numbers is a basic theorem in fuzzy analisys theory, which explains the relationship between fuzzy numbers and interval sets.

**Theorem 1** ([14]) *If $\tilde{u} \in \mathbb{R}_{\mathcal{F}}$, then*

*(1) $[\tilde{u}]_r$ is a closed interval, $r \in [0, 1]$;*

*(2) $[\tilde{u}]_{r_1} \supset [\tilde{u}]_{r_2}$ whenever $0 \leq r_1 \leq r_2 \leq 1$;*

*(3) for any $r_n$ converging increasingly to $r \in (0, 1]$, $\bigcap_{n=1}^{\infty} [\tilde{u}]_{r_n} = [\tilde{u}]_r$.*

*Conversely, if $\{A_r : r \in [0, 1]\}$ fulfills (1)-(3), then there exists a unique $\tilde{u} \in \mathbb{R}_{\mathcal{F}}$ such that $[\tilde{u}]_r = A_r$ for $r \in (0, 1]$ and $[\tilde{u}]_0 = \overline{\bigcup_{r \in (0,1]} [\tilde{u}]_r} \subset A_0$.*

As we all know, the calculus on time scale is the unification of difference equation theory and differential equation theory, and it has applications in any field where discrete and continuous data need to be modeled simultaneously.

**Definition 1** ([37]) *A time scale $\mathbb{T}$ is a nonempty, closed subset of $\mathbb{R}$, equipped with the topology induced from the standard topology on $\mathbb{R}$.*

**Definition 2** ([37]) *The forward(backward) jump operator $\sigma(t)$ at t for $t < \sup\ \mathbb{T}$( respectively, $\rho(t)$ at t for $t > \inf\ \mathbb{T}$) is given by*

$$\sigma(t) = \inf\{\tau > t : \tau \in \mathbb{T}\}, (\rho(t) = \sup\{\tau < t : \tau \in \mathbb{T}\})$$

*where $t \in \mathbb{T}$.*

Additionally, $\sigma(\sup \mathbb{T}) = \sup \mathbb{T}$, if $\sup \mathbb{T} < \infty$, and if $\inf \mathbb{T} > -\infty$ then $\rho(\inf \mathbb{T}) = \inf \mathbb{T}$. Furthermore, the graininess function $\mu : \mathbb{T} \to \mathbb{R}^+$ is defined by $\mu(t) = \sigma(t) - t$ and also the left-graininess function $v : \mathbb{T} \to \mathbb{R}^+$ is defined by $v(t) = t - \rho(t)$.

**Definition 3** ([37]) *If $\sigma(t) > t$, then the point t is called right-scattered; while if $\rho(t) < t$, then t is termed left-scattered. If $t < \sup\ \mathbb{T}$ and $\sigma(t) = t$, then the point t is called right-dense; while if $t > \inf\ \mathbb{T}$ and $\rho(t) = t$, then t is left-dense.*

**Definition 4** ([38]) *A function $\tilde{f} : \mathbb{T} \to \mathbb{R}_{\mathcal{F}}$ is called rd-continuous provided it is continuous at right-dense points in $\mathbb{T}$ and its left-sided limits exist (finite) at left-dense points in $\mathbb{T}$.*

In what follows, unless stated otherwise, we consider a function is continuous, which means the function is *rd*-continuous.

If $\mathbb{T}$ has a left-scattered maximum at $m \in \mathbb{R}$, then we define $\mathbb{T}^k = \mathbb{T} - \{m\}$, otherwise $\mathbb{T}^k = \mathbb{T}$. In summary,

$$\mathbb{T}^k = \begin{cases} \mathbb{T} \setminus (\rho(\sup \mathbb{T}), \sup \mathbb{T}], & \textit{if } \sup \mathbb{T} < \infty, \\ \mathbb{T}, & \textit{if } \sup \mathbb{T} = \infty. \end{cases}$$

Denote $[a, b)_{\mathbb{T}} = \{t \in \mathbb{T} : a \leq t \leq b\}$, then

$$[a, b]_{\mathbb{T}}^k = \begin{cases} [a, b]_{\mathbb{T}}, & \textit{if } a < \rho(b) = b, \\ [a, \rho(b)]_{\mathbb{T}}, & \textit{if } a < \rho(b) < b. \end{cases}$$

## The fuzzy Henstock-Stieltjes Δ-integral on time scales

In this section, we define the fuzzy Henstock-Stieltjes Δ-integral on time scales. In addition, we also obtain some properties and several necessary and sufficient conditions for fuzzy Henstock-Stieltjes Δ-integrability. Next, we will define the fuzzy Henstock-Stieltjes Δ-integral.

Let us first introduce an important auxiliary concept, which is so called Δ-gauge and $\delta$-fine partition of $[a, b]_{\mathbb{T}}$.

**Definition 5** ([27]) *We say* $\delta = (\delta_L, \delta_R)$ *is a* $\Delta$*-gauge for* $[a, b]_{\mathbb{T}}$ *provided* $\delta_L(t) > 0$ *on* $(a, b]_{\mathbb{T}}, \delta_R(t) > 0$ *on* $[a, b)_{\mathbb{T}}, \delta_L(a) \geq 0, \delta_R(b) \geq 0,$ *and* $\delta_R(t) \geq \mu(t)$ *for all* $t \in [a, b)_{\mathbb{T}}$.

**Definition 6** ([27]) *Let P be a partition of* $[a, b]_{\mathbb{T}}$ *denoted by*

$$P = \{a = t_0 \leq \xi_1 \leq t_1 \leq \cdots \leq t_{i-1} \leq \xi_i \leq t_i = b\}$$

*for* $t_i \in [a, b]_{\mathbb{T}}$ *and* $1 \leq i \leq n$. *We shall call each* $\xi_i \in [a, b]_{\mathbb{T}}$ *a tag-point and each* $t_i$ *an endpoint.*

**Definition 7** ([27]) *Let P be* $\delta$*-fine, if* $\delta$ *is a* $\Delta$*-gauge on* $[a, b]_{\mathbb{T}}$, *and for* $i = 1, 2, \cdots, n$ *such that*

$$\xi_i \in [t_{i-1}, t_i] \subset [\xi_i - \delta_L(\xi_i), \xi_i + \delta_R(\xi_i)].$$

With the preparation of $\delta$-fine partition, we can introduce the concept of generalized fuzzy Henstock-Stieltjes $\Delta$-integral. The definition of this integral is different from that given in [16, 31, 39] because the function $\alpha$ in the expression of the integral is a bounded variable function.

**Definition 8** *Let* $\alpha : [a, b]_{\mathbb{T}} \to \mathbb{R}$ *be a function of bounded variation. Let* $\tilde{f} : [a, b]_{\mathbb{T}} \to \mathbb{R}_{\mathcal{F}}$ *is fuzzy Henstock-Stieltjes* $\Delta$*-integrable with respect to* $\alpha$ *on* $[a, b]_{\mathbb{T}}$ *if there exists a* $\tilde{H} \in \mathbb{R}_{\mathcal{F}}$ *with the following property: for every* $\varepsilon > 0$ *there is a* $\Delta$*-gauge* $\delta$ *such that for any* $\delta$*-fine partitions* $P = \{[t_{i-1}, t_i]; \xi_i\}_{i=1}^n$ *of* $[a, b]_{\mathbb{T}}$, *we have*

$$D\left(\sum_{i=1}^n \tilde{f}(\xi_i)(\alpha(t_i) - \alpha(t_{i-1})), \tilde{H}\right) < \varepsilon.$$

*We write*

$$(FHS) \int_a^b \tilde{f}(t)\Delta\alpha = \tilde{H}.$$

The family of all fuzzy Henstock-Stieltjes $\Delta$-integrable functions on $[a, b]_{\mathbb{T}}$ is denoted by $FHS_{[a,b]_{\mathbb{T}}}$.

**Remark 1** *We note that*

*(1) In Definition 8, if* $\delta$ *is a constant, then we call* $\tilde{f}$ *is fuzzy Riemann-Stieltjes* $\Delta$*-integrable with respect to* $\alpha$ *on* $[a, b]_{\mathbb{T}}$;

*(2) In Definition 8, if* $\alpha(t) = t$, *then we call* $\tilde{f}$ *is fuzzy Henstock* $\Delta$*-integrable on* $[a, b]_{\mathbb{T}}$. *Zhao et al.* [39] *have also carried out a detailed study;*

*(3) In Definition 8, if* $\mathbb{T} = [a, b], \alpha(t) = t$, *then we call* $\tilde{f}$ *is fuzzy Henstock integrable on* $[a, b]$. *Wu and Gong have been introduced in detail in* [14, 15].

**Example 1** *Let* $[a, b)_{\mathbb{T}}$ *contains a countable infinite subset* $\bigcup_{i=1}^{\infty}\{\tau_i\}$ *with* $\sigma(\tau_i) = \tau_i$. *Let* $\alpha(t) = t, \tilde{f} : [a, b]_{\mathbb{T}} \to \mathbb{R}_{\mathcal{F}}$ *be given by*

$$\tilde{f}(t) = \begin{cases} \tilde{\theta}, & t = \tau_i, \\ \tilde{0}, & t \neq \tau_i, \end{cases}$$

*where* $\tilde{\theta}$ *is a fuzzy number. Let* $\varepsilon > 0$ *be given, then define a* $\Delta$*-gauge* $\delta$ *on* $[a, b]_{\mathbb{T}}$ *by* $\delta_L(\tau_i) = \delta_R(\tau_i) = \frac{\varepsilon}{2^{i+2}}, i \geq 1, \delta_R(t) = \max\{1, \mu(t)\}$ *and* $\delta_L(t) = 1$ *for* $t \in [a, b]_{\mathbb{T}} \setminus \bigcup_{i=1}^{\infty}\{\tau_i\}$. *Let P be*

*a δ-fine partition of $[a, b]_{\mathbb{T}}$, then*

$$D\left(\sum_{i=1}^{n} \tilde{f}(\xi_i)(\alpha(t_i) - \alpha(t_{i-1})), \tilde{0}\right)$$

$$= D\left(\sum_{i=1}^{n} \tilde{f}(\xi_i)(t_i - t_{i-1}), \tilde{0}\right)$$

$$\leq D\left(\sum_{i=1}^{\infty} \tilde{\theta}(\delta_L(\tau_i) + \delta_R(\tau_i)), \tilde{0}\right)$$

$$= D\left(\sum_{i=1}^{\infty} \frac{\tilde{\theta}\varepsilon}{2^{i+1}}, \tilde{0}\right) < \varepsilon.$$

*Therefore, we get $\tilde{f}$ is fuzzy Henstock Δ-integrable on $[a, b]_{\mathbb{T}}$ and*

$$\int_a^b \tilde{f}(t)\Delta\alpha = \int_a^b \tilde{f}(t)\Delta t = \tilde{0}.$$

**Theorem 2** *Let $\alpha : [a, b]_{\mathbb{T}} \to \mathbb{R}$ be a function of bounded variation. If $\tilde{f} : [a, b]_{\mathbb{T}} \to \mathbb{R}_{\mathcal{F}}$ is FHS-Δ-integrable, then the integral value is unique.*

**Proof.** Assume integral value is not unique and denote $\tilde{A} = \int_a^b \tilde{f}(t)\Delta\alpha$ and $\tilde{B} = \int_a^b \tilde{f}(t)\Delta\alpha$. Let $\varepsilon > 0$ be given. Then there is a Δ-gauge $\delta$ of $[a, b]_{\mathbb{T}}$ such that for all $\delta$-fine partition of $[a, b]_{\mathbb{T}}$, we have

$$D\left(\sum_{i=1}^{n} \tilde{f}(\xi_i)(\alpha(t_i) - \alpha(t_{i-1})), \tilde{A}\right) < \frac{\varepsilon}{2},$$

$$D\left(\sum_{i=1}^{n} \tilde{f}(\xi_i)(\alpha(t_i) - \alpha(t_{i-1})), \tilde{B}\right) < \frac{\varepsilon}{2}.$$

It follows that

$$
\begin{aligned}
D(\tilde{A}, \tilde{B}) &\leq D\left(\sum_{i=1}^{n} \tilde{f}(\xi_i)(\alpha(t_i) - \alpha(t_{i-1})), \tilde{A}\right) \\
&\quad + D\left(\sum_{i=1}^{n} \tilde{f}(\xi_i)(\alpha(t_i) - \alpha(t_{i-1})), \tilde{B}\right) \\
&< \frac{\varepsilon}{2} + \frac{\varepsilon}{2} = \varepsilon.
\end{aligned}
$$

Consequently, the integral value is unique. The proof is complete.

**Theorem 3** *Let $\alpha : [a, b]_{\mathbb{T}} \to \mathbb{R}$ be a function of bounded variation. If $\tilde{f} : [a, b]_{\mathbb{T}} \to \mathbb{R}_{\mathcal{F}}$ is FHS-Δ-integral with respect to $\alpha$ on $[a, b]_{\mathbb{T}}$, then $\tilde{f}$ is also FHS-Δ-integral with respect to $\alpha$ on every subinterval of $[a, b]_{\mathbb{T}}$.*

**Proof.** Let $\varepsilon > 0$ and $[c, d]_{\mathbb{T}} \subseteq [a, b]_{\mathbb{T}}$. Since $\tilde{f} : [a, b]_{\mathbb{T}} \to \mathbb{R}_{\mathcal{F}}$ is FHS-Δ-integral with respect to $\alpha$ on $[a, b]_{\mathbb{T}}$, there exists a Δ-gauge $\delta$ on $[a, b]_{\mathbb{T}}$ such that for $\delta$-fine partition $P_1$ and $P_2$ of

$[a, b]_{\mathbb{T}}$, we have

$$D\left(\sum_{P_1} \tilde{f}(\xi)(\alpha(v) - \alpha(u)), \sum_{P_2} \tilde{f}(\xi')(\alpha(v') - \alpha(u'))\right) < \varepsilon.$$

Let $P_a$ and $P_b$ be the divisions of $[a, c]_{\mathbb{T}}$ and $[d, b]_{\mathbb{T}}$, respectively. Let $P'_1$ and $P'_2$ be partitions of $[c, d]_{\mathbb{T}}$ and define $P_1 = P_a \cup P'_1 \cup P_b$ and $P_2 = P_a \cup P'_2 \cup P_b$. Then $P_1$ and $P_2$ are partitions of $[a, b]_{\mathbb{T}}$ which are $\delta$-fine and

$$
\begin{aligned}
&D\left(\sum_{P'_1} \tilde{f}(\xi)(\alpha(v) - \alpha(u)), \sum_{P'_2} \tilde{f}(\xi')(\alpha(v') - \alpha(u'))\right) \\
=\ &D\left(\sum_{P_1} \tilde{f}(\xi)(\alpha(v) - \alpha(u)), \sum_{P_2} \tilde{f}(\xi')(\alpha(v') - \alpha(u'))\right) \\
<\ &\varepsilon.
\end{aligned}
$$

Hence, by Theorem 5, the function $\tilde{f}$ is FHS-$\Delta$-integral with respect to $\alpha$ on every subinterval of $[a, b]_{\mathbb{T}}$. The proof is complete.

We note that we can know that fuzzy Henstock-Stieltjes $\Delta$-integral is the generalization of fuzzy Riemann-Stieltjes $\Delta$-integral.

**Theorem 4** *Let $\alpha$ be a function of bounded variation on $[a, b]_{\mathbb{T}}$. $\tilde{f} : [a, b]_{\mathbb{T}} \to \mathbb{R}_{\mathcal{F}}$ is fuzzy Riemann-Stieltjes $\Delta$-integrable with respect to $\alpha$ on $[a, b]_{\mathbb{T}}$. Then $(\tilde{f}, \alpha) \in FHS_{[a,b]_{\mathbb{T}}}$ and*

$$(FRS)\int_a^b \tilde{f}(t)\Delta\alpha = (FHS)\int_a^b \tilde{f}(t)\Delta\alpha.$$

**Proof.** Since the proof of Theorem 4 is straightforward following the pattern of [16]. Therefore, we omitted the proof.

**Example 2** *Let $\mathbb{T} = \left[\frac{1}{2^{i+1}}, \frac{1}{2^i}\right]$, $i$ is odd. Let $\alpha$ be a decreasing function, $\tilde{f} : [0, 1] \bigcap \mathbb{T} \to \mathbb{R}_{\mathcal{F}}$ be given by $\tilde{f}(t)(s) = \Theta(t)\tilde{u}(s)$, where*

$$\tilde{u}(s) = \begin{cases} s, & 0 \le s \le 1, \\ 2 - s, & 1 < s \le 2, \\ 0, & \text{otherwise} \end{cases} \Theta(t) = \begin{cases} 1, t \text{ is irational,} \\ 0, t \text{ is irrational.} \end{cases}$$

*For any $\varepsilon > 0$, we define*

$$\delta(t) = \begin{cases} \frac{1}{2}, & t \text{ is irrational,} \\ \frac{1}{2}\min\{|v(t)|, |\mu(t)|\}, & t \text{ is irational,} \end{cases}$$

*then for any $\delta$-fine partition $P = \{[t_{i-1}, t_i]; \xi_i\}$, we have*

$$D\left(\sum_P \tilde{f}(\xi_i)(s)(\alpha(t_i) - \alpha(t_{i-1})), \tilde{0}\right) < \varepsilon.$$

*Therefore, $\tilde{f}$ is FHS-$\Delta$-integrable with respect to $\alpha$ on $[0, 1]_{\mathbb{T}}$. In addition, if all the associated points $\xi_i$ are rational in $[0, 1]_{\mathbb{T}}$, then fuzzy Riemann-Stieltjes sums*

$$\sum_{i=1}^{n} \Theta(\xi_i) \tilde{u}(\alpha(t_i) - \alpha(t_{i-1})) = \tilde{u}\alpha([0, 1]_{\mathbb{T}}),$$

*where for each $A = [a_1, a_2]$, $B = [b_1, b_2]$ with $A \cap B = \emptyset$, $\alpha(A \cup B) = \alpha(a_2) - \alpha(a_1) + \alpha(b_2) - \alpha(b_1)$, since $\mathbb{T}$ is a collection of closed intervals, then $\alpha([0, 1]_{\mathbb{T}})$ is well defined. If all the associated points $\xi_i$ are irrational in $[0, 1]_{\mathbb{T}}$, then fuzzy Riemann-Stieltjes sums*

$$\sum_{i=1}^{n} \Theta(\xi_i) \tilde{u}(\alpha(t_i) - \alpha(t_{i-1})) = \tilde{0}.$$

*Thus, $\tilde{f}$ is not FRS-$\Delta$-integrable with respect to $\alpha$ on $[0, 1]_{\mathbb{T}}$.*

In order to discuss the properties of FHS-$\Delta$-integrals, we first give a Chauchy's criterion for the existence of this integrals.

**Theorem 5**. *Let $\alpha : [a, b]_{\mathbb{T}} \to \mathbb{R}$ be a function of bounded variation. If $\tilde{f} : [a, b]_{\mathbb{T}} \to \mathbb{R}_{\mathcal{F}}$ is FHS-$\Delta$-integrable with respect to $\alpha$ on $[a, b]_{\mathbb{T}}$ if and only if for every $\varepsilon > 0$, there is a $\Delta$-gauge $\delta$ on $[a, b]_{\mathbb{T}}$ such that for $\delta$-fine partitions $P_1 = \{[u, v]; \xi\}$ and $P_2 = \{[u', v']; \xi'\}$ we have*

$$D\left(\sum_{P_1} \tilde{f}(\xi)(\alpha(v) - \alpha(u)), \sum_{P_2} \tilde{f}(\xi')(\alpha(v') - \alpha(u'))\right) < \varepsilon.$$

**Proof.** Since $\tilde{f} : [a, b]_{\mathbb{T}} \to \mathbb{R}_{\mathcal{F}}$ is FHS-$\Delta$-integrable with respect to $\alpha$ on $[a, b]_{\mathbb{T}}$, let $\tilde{H} = (FHS) \int_a^b \tilde{f}(t) \Delta\alpha$, for every $\varepsilon > 0$, there is a $\Delta$-gauge $\delta_1$ such that for $\delta_1$-fine partition $P_1 = \{[u, v]; \xi\}$ we have

$$D\left(\sum_{P_1} \tilde{f}(\xi)(\alpha(v) - \alpha(u)), \tilde{H}\right) < \frac{\varepsilon}{2}.$$

Similarly, there is a $\Delta$-gauge $\delta_2$ such that for $\delta_2$-fine partition $P_2 = \{[u', v']; \xi'\}$ we have

$$D\left(\sum_{P_2} \tilde{f}(\xi')(\alpha(v') - \alpha(u')), \tilde{H}\right) < \frac{\varepsilon}{2}.$$

Then

$$D\left(\sum_{P_1} \tilde{f}(\xi)(\alpha(v) - \alpha(u)), \sum_{P_2} \tilde{f}(\xi')(\alpha(v') - \alpha(u'))\right)$$
$$\leq D\left(\sum_{P_1} \tilde{f}(\xi)(\alpha(v) - \alpha(u)), \tilde{H}\right) + D\left(\sum_{P_2} \tilde{f}(\xi')(\alpha(v') - \alpha(u')), \tilde{H}\right)$$
$$< \frac{\varepsilon}{2} + \frac{\varepsilon}{2} < \varepsilon.$$

Therefore

$$D\left(\sum_{P_1}\tilde{f}(\xi)(\alpha(v)-\alpha(u)),\sum_{P_2}\tilde{f}(\xi')(\alpha(v')-\alpha(u'))\right)<\varepsilon.$$

Conversely, for each positive integer $n$, there exists a $\Delta$-gauge $\delta_n$ on $[a,b]_{\mathbb{T}}$ such that for $\delta_n$-fine partitions $P_1=\{[u,v];\xi\}$ and $P_2=\{[u',v'];\xi'\}$ of $[a,b]_{\mathbb{T}}$ we have

$$D\left(\sum_{P_1}\tilde{f}(\xi)(\alpha(v)-\alpha(u)),\sum_{P_2}\tilde{f}(\xi')(\alpha(v')-\alpha(u'))\right)<\frac{1}{n}.$$

Now, we assume that the sequence $\{\delta_n\}$ is nonincreasing. For each $n$, let $P_n$ and $P_m$ be $\delta_n$-fine and $\delta_m$-fine partitions of $[a,b]_{\mathbb{T}}$, respectively. Then

$$D\left(\sum_{P_n}\tilde{f}(\xi)(\alpha(v)-\alpha(u)),\sum_{P_m}\tilde{f}(\xi')(\alpha(v')-\alpha(u'))\right)<\frac{1}{N}$$

for $m,n>N,N\in\mathbb{N}$.

Therefore, the sequence $\{\sum_{P_n}\tilde{f}(\xi)(\alpha(v)-\alpha(u))\}$ is a Cauchy sequence in $\mathbb{R}_{\mathcal{F}}$.

Let $\tilde{H}$ be the limit of that sequence and $\varepsilon>0$ be given. Choose a positive integer N such that $\frac{1}{N}<\frac{\varepsilon}{2}$ and

$$D\left(\sum_{P_n}\tilde{f}(\xi)(\alpha(v)-\alpha(u)),\tilde{H}\right)<\frac{\varepsilon}{2},\forall\,n>N.$$

Let $P_N$ be a $\delta_N$-fine partition of $[a,b]_{\mathbb{T}}$, then

$$D\left(\sum_{P}\tilde{f}(\xi)(\alpha(v)-\alpha(u)),\tilde{H}\right)$$

$$\leq\quad D\left(\sum_{P}\tilde{f}(\xi)(\alpha(v)-\alpha(u)),\sum_{P_N}\tilde{f}(\xi)(\alpha(v)-\alpha(u))\right)$$

$$+D\left(\sum_{P_N}\tilde{f}(\xi)(\alpha(v)-\alpha(u)),\tilde{H}\right)$$

$$<\quad\frac{1}{N}+\frac{\varepsilon}{2}$$

$$<\quad\frac{\varepsilon}{2}+\frac{\varepsilon}{2}=\varepsilon.$$

Consequently, $D\left(\sum_{P}\tilde{f}(\xi)(\alpha(v)-\alpha(u)),\tilde{H}\right)<\varepsilon.$ This shows that the function $\tilde{f}$ :

$[a,b]_{\mathbb{T}}\to\mathbb{R}_{\mathcal{F}}$ is FHS-$\Delta$-integrable with respect to $\alpha$ on $[a,b]_{\mathbb{T}}$. The proof is complete.

**Remark 2** *Let* $\alpha:[a,b]\to\mathbb{R}$, *then the result is the same as* $\alpha:[a,b]_{\mathbb{T}}\to\mathbb{R}$.

As with the properties of integrals in real analysis, we give the basic properties of integrals, including interval additivity and linearity.

**Theorem 6** *Let* $\alpha : [a, b]_{\mathbb{T}} \to \mathbb{R}$ *be a function of bounded variation. Let* $\tilde{f} : [a, b]_{\mathbb{T}} \to \mathbb{R}_{\mathcal{F}}$ *and* $c \in [a, b]_{\mathbb{T}}$. *If* $\tilde{f}$ *is FHS-$\Delta$-integrable with respect to* $\alpha$ *on* $[a, c]_{\mathbb{T}}$ *and* $[c, b]_{\mathbb{T}}$, *then* $\tilde{f}$ *is FHS-$\Delta$-integrable with respect to* $\alpha$ *on* $[a, b]_{\mathbb{T}}$ *and*

$$\int_a^b \tilde{f}(t)\Delta\alpha = \int_a^c \tilde{f}(t)\Delta\alpha + \int_c^b \tilde{f}(t)\Delta\alpha.$$

**Proof.** Since $\tilde{f}$ is FHS-$\Delta$-integrable with respect to $\alpha$ on each of the subintervals $[a, c]_{\mathbb{T}}$ and $[c, b]_{\mathbb{T}}$ of $[a, b]_{\mathbb{T}}$, let $\varepsilon > 0$ be given. By hypothesis, there exists a $\Delta$-gauge $\delta_1$ on $[a, c]_{\mathbb{T}}$ such that for $\delta_1$-fine partition $P_1$ of $[a, c]_{\mathbb{T}}$ and $\delta_2$-fine partition $P_2$ of $[c, b]_{\mathbb{T}}$, we have

$$D\left( \sum_{P_1} \tilde{f}(\xi)(\alpha(v) - \alpha(u)), \int_a^c \tilde{f}\Delta\alpha \right) < \frac{\varepsilon}{2},$$

$$D\left( \sum_{P_2} \tilde{f}(\xi)(\alpha(v) - \alpha(u)), \int_c^b \tilde{f}\Delta\alpha \right) < \frac{\varepsilon}{2}.$$

We define a $\Delta$-gauge $\delta = (\delta_L, \delta_R)$ on $[a, b]_{\mathbb{T}}$ by

$$\delta_L(t) = \begin{cases} \delta_L^1(t), & \text{if } t \in [a, c)_{\mathbb{T}}, \\[2mm] \delta_L^1(t), & \text{if } v(t) = 0, \\[2mm] min\{\delta_L^1(t), \frac{v(t)}{2}\}, & \text{if } v(t) > 0, \\[2mm] min\{\delta_L^2(t), \frac{t-c}{2}\}, & \text{if } t \in (c, b]_{\mathbb{T}}, \end{cases}$$

and

$$\delta_R(t) = \begin{cases} min\{\delta_R^1(t), max\{\mu(t), \frac{c-t}{2}\}\}, & \text{if } t \in [a, c)_{\mathbb{T}}, \\[2mm] \delta_R^2(t), & \text{if } t \in [c, b]_{\mathbb{T}}. \end{cases}$$

Let $P' = \{[t'_{i-1}, t'_i]; \xi'_i\}_{i=1}^p$ be a $\delta$-fine partition of $[a, b]_{\mathbb{T}}$. It follows that
case(1) $c = \xi'_k$ and $t'_k > c$;
case(2) $\xi'_k = \rho(c) < c$ and $t'_k = c$.

In case(2), we can draw a conclusion easily. In the first case,

$$D\left(\sum_{i=1}^{p} \tilde{f}(\xi_i')(\alpha(t_i') - \alpha(t_{i-1}')), \int_a^c \tilde{f}\Delta\alpha + \int_c^b \tilde{f}\Delta\alpha\right)$$

$$= D\left(\sum_{i=1}^{k-1} \tilde{f}(\xi_i')(\alpha(t_i') - \alpha(t_{i-1}')) + \tilde{f}(c)(c - t_{k-1}')\right.$$

$$\left. + \tilde{f}(c)(t_k' - c) + \sum_{i=k+1}^{p} \tilde{f}(\xi_i')(\alpha(t_i') - \alpha(t_{i-1}')), \int_a^c \tilde{f}\Delta\alpha + \int_c^b \tilde{f}\Delta\alpha\right)$$

$$\leq D\left(\sum_{i=1}^{k-1} \tilde{f}(\xi_i')(\alpha(t_i') - \alpha(t_{i-1}')) + \tilde{f}(c)(c - t_{k-1}'), \int_a^c \tilde{f}\Delta\alpha\right)$$

$$+ D\left(\sum_{i=k+1}^{p} \tilde{f}(\xi_i')(\alpha(t_i') - \alpha(t_{i-1}')) + \tilde{f}(c)(t_k' - c), \int_c^b \tilde{f}\Delta\alpha\right)$$

$$< \frac{\varepsilon}{2} + \frac{\varepsilon}{2} = \varepsilon.$$

Therefore for given $\varepsilon > 0$, there is a $\Delta$-gauge $\delta$ on $[a, b]_{\mathbb{T}}$ such that for $\delta$-fine partition of $[a, b]_{\mathbb{T}}$ we have

$$D\left(\sum_P \tilde{f}(\xi)(\alpha(v) - \alpha(u)), \int_a^c \tilde{f}\Delta\alpha + \int_c^b \tilde{f}\Delta\alpha\right) < \varepsilon.$$

This shows that $\tilde{f}$ is FHS-$\Delta$-integrable with respect to $\alpha$ on $[a, b]_{\mathbb{T}}$, and

$$\int_a^b \tilde{f}(t)\Delta\alpha = \int_a^c \tilde{f}(t)\Delta\alpha + \int_c^b \tilde{f}(t)\Delta\alpha.$$

The proof is complete.

**Theorem 7** *Let $\alpha : [a, b]_{\mathbb{T}} \to \mathbb{R}$ be a function of bounded variation, $\tilde{f}, \tilde{g} : [a, b]_{\mathbb{T}} \to \mathbb{R}_{\mathcal{F}}$ be FHS-$\Delta$-integrable with respect to $\alpha$ on $[a, b]_{\mathbb{T}}$. Then*

*(1) $k\tilde{f}$ is FHS-$\Delta$-integrable with respect to $\alpha$ on $[a, b]_{\mathbb{T}}$ and*

$$\int_a^b k\tilde{f}(t)\Delta\alpha = k \int_a^b \tilde{f}(t)\Delta\alpha,$$

*where $k \in \mathbb{R}$.*

*(2) $\tilde{f} + \tilde{g}$ is FHS-$\Delta$-integrable with respect to $\alpha$ on $[a, b]_{\mathbb{T}}$ and*

$$\int_a^b (\tilde{f}(t) + \tilde{g}(t))\Delta\alpha = \int_a^b \tilde{f}(t)\Delta\alpha + \int_a^b \tilde{g}(t)\Delta\alpha.$$

**Proof.** (1) Let $\tilde{f} : [a, b]_{\mathbb{T}} \to \mathbb{R}_{\mathcal{F}}$ be FHS-$\Delta$-integrable with respect to $\alpha$ on $[a, b]_{\mathbb{T}}$.
Case 1. If $k = 0$, then the result is obvious.

Case 2. If $k \neq 0$. For given $\varepsilon > 0$, there exists a $\Delta$-gauge $\delta$ on $[a, b]_{\mathbb{T}}$ such that for $\delta$-fine partition $P$ of $[a, b]_{\mathbb{T}}$, we have

$$D\left(\sum_{P} \tilde{f}(\xi)(\alpha(v) - \alpha(u)), \int_a^b \tilde{f}\Delta\alpha\right) < \frac{\varepsilon}{|k|}.$$

Therefore,

$$|k| \cdot D\left(\sum_{P} \tilde{f}(\xi)(\alpha(v) - \alpha(u)), \int_a^b \tilde{f}\Delta\alpha\right) < \varepsilon,$$

that is

$$D\left(\sum_{P} k\tilde{f}(\xi)(\alpha(v) - \alpha(u)), k\int_a^b \tilde{f}\Delta\alpha\right) < \varepsilon.$$

This implies that the function $k\tilde{f}$ is FHS-$\Delta$-integrable with respect to $\alpha$ on $[a, b]_{\mathbb{T}}$ and

$$\int_a^b k\tilde{f}(t)\Delta\alpha = k\int_a^b \tilde{f}(t)\Delta\alpha.$$

(2) Let $\tilde{f}, \tilde{g} : [a, b]_{\mathbb{T}} \to \mathbb{R}_{\mathcal{F}}$ be FHS-$\Delta$-integrable with respect to $\alpha$ on $[a, b]_{\mathbb{T}}$. Then there exists a $\Delta$-gauge $\delta$ on $[a, b]_{\mathbb{T}}$ such that for $\delta$-fine partition $P_1 = \{[u, v], \xi\}$ and $P_2 = \{[u', v'], \xi'\}$ of $[a, b]_{\mathbb{T}}$, we have

$$D\left(\sum_{P_1} \tilde{f}(\xi)(\alpha(v) - \alpha(u)), \int_a^b \tilde{f}\Delta\alpha\right) < \frac{\varepsilon}{2},$$

$$D\left(\sum_{P_2} \tilde{g}(\xi')(\alpha(v') - \alpha(u')), \int_a^b \tilde{g}\Delta\alpha\right) < \frac{\varepsilon}{2},$$

due to FHS-$\Delta$-integrability of $\tilde{f}, \tilde{g}$ and show the validity of the result in this case. Consequently, $\tilde{f} + \tilde{g}$ is FHS-$\Delta$-integrable with respect to $\alpha$ on $[a, b]_{\mathbb{T}}$. The proof is complete.

Next, we give the definition of bounded variation for fuzzy number valued functions on $[a, b]_{\mathbb{T}}$.

**Definition 9** *Given P*: $a = t_0 \leq t_1 \leq \cdots \leq t_{i-1} \leq t_i = b$, *a function* $f : [a, b]_{\mathbb{T}} \to \mathbb{R}$ *is said to be of bounded variation on* $[a, b]_{\mathbb{T}}$ *if*

$$Var(f, [a, b]_{\mathbb{T}}) = \sup_{P} \sum_{i=1}^{n} |f(t_i) - f(t_{i-1})| < \infty.$$

$Var(f, [a, b]_{\mathbb{T}})$ *is called the total variation of f on* $[a, b]_{\mathbb{T}}$.

We note that the function of bounded variation is defined on the set of all real numbers [12], the above definition is a special case.

**Theorem 8** *Let $\alpha : [a, b]_\mathbb{T} \to \mathbb{R}$ be a function of bounded variation. If $\tilde{f} : [a, b]_\mathbb{T} \to \mathbb{R}_\mathcal{F}$ is FHS-$\Delta$-integrable with respect to $\alpha$ on $[a, b]_\mathbb{T}$, then*

$$\| \int_a^b \tilde{f}\Delta\alpha \| \leq \sup_{t \in [a,b]_\mathbb{T}} \| \tilde{f}(t) \| \cdot Var(\alpha, [a, b]_\mathbb{T}).$$

**Proof.** Since $\tilde{f} : [a, b]_\mathbb{T} \to \mathbb{R}_\mathcal{F}$ is FHS-$\Delta$-integrable with respect to $\alpha$ on $[a, b]_\mathbb{T}$, let $\varepsilon > 0$ be given, by assumption, there exists a $\Delta$-gauge $\delta$ of $[a, b]_\mathbb{T}$ such that for any $\delta$-fine partitions $P = \{[t_{i-1}, t_i]; \xi_i\}_{i=1}^n$ of $[a, b]_\mathbb{T}$, we have

$$D\left( \sum_{i=1}^n \tilde{f}(\xi_i)(\alpha(t_i) - \alpha(t_{i-1})), \int_a^b \tilde{f}\Delta\alpha \right) < \varepsilon.$$

Then

$$
\begin{aligned}
\| \int_a^b \tilde{f}\Delta\alpha \| \quad &\leq \quad D\left( \int_a^b \tilde{f}\Delta\alpha, \sum_{i=1}^n \tilde{f}(\xi_i)(\alpha(t_i) - \alpha(t_{i-1})) \right) \\
&\quad + D\left( \sum_{i=1}^n \tilde{f}(\xi_i)(\alpha(t_i) - \alpha(t_{i-1})), \tilde{0} \right) \\
&< \quad \varepsilon + \sum_{i=1}^n \| \tilde{f}(\xi_i) \| \cdot | \alpha(t_i) - \alpha(t_{i-1}) | \\
&\leq \quad \varepsilon + \sup_{t \in [a,b]_\mathbb{T}} \| \tilde{f}(t) \| \cdot Var(\alpha, [a, b]_\mathbb{T}).
\end{aligned}
$$

Since $\varepsilon > 0$ is arbitrary, we have

$$\| \int_a^b \tilde{f}\Delta\alpha \| \leq \sup_{t \in [a,b]_\mathbb{T}} \| \tilde{f}(t) \| \cdot Var(\alpha, [a, b]_\mathbb{T}).$$

The proof is complete.

We note that above theorem gives the best estimate of FHS-$\Delta$-integrals.

**Theorem 9** *Let $\alpha : [a, b]_\mathbb{T} \to \mathbb{R}$ be a function of bounded variation. If $\tilde{f}(t) = \tilde{g}(t)$ almost everywhere on $[a, b]_\mathbb{T}$ and $\tilde{f}$ is FHS-$\Delta$-integrable with respect to $\alpha$ on $[a, b]_\mathbb{T}$, then $\tilde{g}$ is FHS-$\Delta$-integrable with respect to $\alpha$ on $[a, b]_\mathbb{T}$ and*

$$\int_a^b \tilde{f}(t)\Delta\alpha = \int_a^b \tilde{g}(t)\Delta\alpha.$$

**Proof.** Since $\tilde{f}(t) : [a, b]_\mathbb{T} \to \mathbb{R}_\mathcal{F}$ is FHS-$\Delta$-integrable on $[a, b]_\mathbb{T}$, we can assume

$$\tilde{H} = \int_a^b \tilde{f}(t)\Delta\alpha.$$

Given $\varepsilon > 0$, there is a $\Delta$-gauge $\delta_0$ such that for any $\delta_0$-fine partition of $[a, b]_\mathbb{T}$, we have

$$D\left( \sum_P \tilde{f}(\xi)(\alpha(v) - \alpha(u)), \tilde{H} \right) < \varepsilon.$$

Set $X = \sum\limits_{i=1}^{\infty} X_i$, where $X_i = \{t : i - 1 < D(\tilde{f}(t), \tilde{g}(t)) \leq i, t \in [a, b]_{\mathbb{T}}\}$. For each $i$ there is $G_i$ which is union of a countable number of open intervals with the total length less than $\varepsilon 2^{-i} i^{-1}$ and such that $X_i \subset G_i \cap \mathbb{T}$. Then, we define

$$\delta(\xi) = \begin{cases} \delta_0(\xi), & \xi \in [a, b]_{\mathbb{T}} \setminus X, \\ \delta(\xi), & (\xi - \delta_L(\xi), \xi + \delta_R(\xi)) \subset G_i \; for \; i = 1, 2, \cdots, \xi \in X_i, \end{cases}$$

such that for any $\delta$-fine partition $P = \{[u, v]; \xi\}$, we have

$$D\left(\sum_P \tilde{g}(\xi)(\alpha(v) - \alpha(u)), \tilde{H}\right)$$

$$= D\left((\sum_{\xi \in X} + \sum_{\xi \notin X})\tilde{g}(\xi)(\alpha(v) - \alpha(u)), \tilde{H}\right)$$

$$= D\left(\sum_{\xi \in X} \tilde{g}(\xi)(\alpha(v) - \alpha(u)) + \sum_{\xi \notin X} \tilde{f}(\xi)(\alpha(v) - \alpha(u)), \tilde{H}\right)$$

$$= D\left(\sum_{\xi \in X} \tilde{g}(\xi)(\alpha(v) - \alpha(u)) + \sum_{\xi \notin X} \tilde{f}(\xi)(\alpha(v) - \alpha(u)) + \sum_{\xi \in X} \tilde{f}(\xi)(\alpha(v) - \alpha(u)),\right.$$

$$\left. \sum_{\xi \in X} \tilde{f}(\xi)(\alpha(v) - \alpha(u)) + \tilde{H}\right)$$

$$\leq D\left(\sum_P \tilde{f}(\xi)(\alpha(v) - \alpha(u)), \tilde{H}\right) + D\left(\sum_{\xi \in X} \tilde{g}(\xi)(\alpha(v) - \alpha(u)),\right.$$

$$\left. \sum_{\xi \in X} \tilde{f}(\xi)(\alpha(v) - \alpha(u))\right)$$

$$\leq \varepsilon + \sum_{i=1}^{\infty} \sum_{\xi \in X} D(\tilde{g}(\xi), \tilde{f}(\xi)) \cdot \mid \alpha(v) - \alpha(u) \mid \leq 2\varepsilon.$$

Consequently, $\tilde{g}$ is FHS-$\Delta$-integrable on $[a, b]_{\mathbb{T}}$, and

$$\int_a^b \tilde{f}(t)\Delta\alpha = \int_a^b \tilde{g}(t)\Delta\alpha.$$

The proof is complete.

## The characterization of FHS-Δ-integrable functions

In this section, we can embed fuzzy number space using a theorem of Goetschel and Voxman [40], namely the set of all fuzzy numbers on $\mathbb{R}$ with the generalized Hausdorff metric, into a concrete Banach space $\overline{C}[0, 1] \times \overline{C}[0, 1]$ (refer to [41]). Here $\overline{C}[0, 1]$ stands for the class of all real-valued bounded functions $f$ on $[0, 1]$ such that $f$ is left continuous for any $x \in (0, 1]$ and $f$ has a right limit for any $x \in [0, 1)$, especially $f$ is right continuous at 0.

**Definition 10**. *Let $X$ be a Banach space. Let $\alpha : [a, b]_{\mathbb{T}} \to \mathbb{R}$ be a function of bounded variation, $f : [a, b]_{\mathbb{T}} \to X$ be a vector-valued function. $f(t)$ be said to be HS-$\Delta$-integrable with respect to $\alpha$ on $[a, b]_{\mathbb{T}}$ if for every $\varepsilon > 0$ there is a $\Delta$-gauge $\delta$ such that for any $\delta$-fine partitions $P = \{[u,$*

*v*]; ξ} *of* $[a, b]_{\mathbb{T}}$, *we have*

$$\| \sum f(\xi)(\alpha(v) - \alpha(u)) - A \| < \varepsilon$$

*where the sum* Σ *is understood to be over P and* ∥·∥ *stands for the norm of X.*

We write (VHS) $\int_a^b f(t)\Delta\alpha = A$ and $f(t) \in X$. Here (VHS) stands for the HS-Δ-integral for vector-valued functions.

In 1991, Wu and Ma [41] shown that the fuzzy number Spaces could be isometrically embedded into a concrete Banach Spaces $\overline{C}[0, 1] \times \overline{C}[0, 1]$ and used as a tool to analyze some properties of fuzzy number valued functions.

**Theorem 10** ([41]) *For* $\tilde{u} \in \mathbb{R}_{\mathcal{F}}$, *denote* $j \circ \tilde{u} = (u^-, u^+)$. *Then* $j(\mathbb{R}_{\mathcal{F}})$ *is a closed convex cone with vertex 0 in* $\overline{C}[0, 1] \times \overline{C}[0, 1]$ *and* $j : \mathbb{R}_{\mathcal{F}} \to \overline{C}[0, 1] \times \overline{C}[0, 1]$ *satisfies*

*(1) for all* $\tilde{u}, \tilde{v} \in \mathbb{R}_{\mathcal{F}}, s \geq 0, t \geq 0, j \circ (s\tilde{u} + t\tilde{v}) = s \cdot j \circ \tilde{u} + t \cdot j \circ \tilde{v}$,

*(2)* $D(\tilde{u}, \tilde{v}) = ||j \circ \tilde{u} - j \circ \tilde{v}||_B$,

*i.e. j embeds* $\mathbb{R}_{\mathcal{F}}$ *into* $\overline{C}[0, 1] \times \overline{C}[0, 1]$ *isometrically and isomorphically, where* ∥·∥_B *represents the norm in* $\overline{C}[0, 1] \times \overline{C}[0, 1]$.

**Theorem 11** *Let* α *be a function of bounded variation on* $[a, b]_{\mathbb{T}}$. *If* $\tilde{f} : [a, b]_{\mathbb{T}} \to \mathbb{R}_{\mathcal{F}}$ *is FHS-Δ-integrable with respect to* α *on* $[a, b]_{\mathbb{T}}$, *then* $j \circ \tilde{f} : [a, b]_{\mathbb{T}} \to \overline{C}[0, 1] \times \overline{C}[0, 1]$ *is Henstock-Stieltjes Δ-integrable with respect to* α *on* $[a, b]_{\mathbb{T}}$ *and*

$$\int_a^b j \circ \tilde{f}\Delta\alpha = j \circ \int_a^b \tilde{f}\Delta\alpha$$

*where j∘ is isometric isomorphic embedding operator defined in Theorem 10.*

**Proof.** Since $\tilde{f} : [a, b]_{\mathbb{T}} \to \mathbb{R}_{\mathcal{F}}$ is FHS-Δ-integrable with respect to α on $[a, b]_{\mathbb{T}}$, let ε > 0 be given, there exists a Δ-gauge δ such that for δ-fine partition P = {[u, v]; ξ} we have

$$D\left( \sum \tilde{f}(\xi)(\alpha(v) - \alpha(u)), \int_a^b \tilde{f}\Delta\alpha \right) < \varepsilon.$$

Hence for $j : \mathbb{R}_{\mathcal{F}} \to \overline{C}[0, 1] \times \overline{C}[0, 1]$, we have

$$\| \sum (j \circ \tilde{f})(\xi)(\alpha(v) - \alpha(u)) - j \circ \int_a^b \tilde{f}\Delta\alpha \|_B$$
$$= D\left( \sum \tilde{f}(\xi)(\alpha(v) - \alpha(u)), \int_a^b \tilde{f}\Delta\alpha \right)$$
$$< \varepsilon.$$

Therefore there exists a Δ-gauge δ on $[a, b]_{\mathbb{T}}$ such that

$$\| \sum (j \circ \tilde{f})(\xi)(\alpha(v) - \alpha(u)) - j \circ \int_a^b \tilde{f}\Delta\alpha \|_B < \varepsilon$$

whenever P is δ-fine partition of $[a, b]_{\mathbb{T}}$.

Hence the function $j \circ \tilde{f}$ is Henstock-Stieltjes Δ-integrable with respect to α on $[a, b]_{\mathbb{T}}$. The proof is complete.

Next, we give the main outcomes of this section. We can give a characterization theorem for FHS-Δ-integrable functions by means of the study of the VHS-Δ-integral on Banach space.

**Theorem 12** *Let $\alpha$ be a function of bounded variation on $[a, b]_{\mathbb{T}}$. If $\tilde{f} : [a, b]_{\mathbb{T}} \to \mathbb{R}_{\mathcal{F}}$, then the following statements are equivalent:*

*(1) $\tilde{f}(t)$ is FHS-$\Delta$-integral with respect to $\alpha$,*

*(2) $g_r^-(t), g_r^+(t)$ are HS-$\Delta$-integrable with respect to $\alpha$ for any $r \in [0, 1]$ uniformly, i.e., for every $\varepsilon > 0$, there is $\Delta$-gauge $\delta$ such that for any $\delta$-fine partition $P = \{[u, v]; \xi\}$ we have*

$$| \sum g_r^-(\xi)(\alpha(v) - \alpha(u)) - H_r^- | < \varepsilon,$$

$$| \sum g_r^+(\xi)(\alpha(v) - \alpha(u)) - H_r^+ | < \varepsilon$$

*where $r \in [0, 1]$ and $H_r^- = \int_a^b g_r^-(t)\Delta\alpha, H_r^+ = \int_a^b g_r^+(t)\Delta\alpha$,*

$$g_r^-(\xi) = \begin{cases} f_r^-(\xi), & \alpha(\sigma(\xi)) > \alpha(\rho(\xi)) \\ \\ f_r^+(\xi), & \alpha(\sigma(\xi)) \leq \alpha(\rho(\xi)), \end{cases} \quad g_r^+(\xi) = \begin{cases} f_r^+(\xi), & \alpha(\sigma(\xi)) > \alpha(\rho(\xi)) \\ \\ f_r^-(\xi), & \alpha(\sigma(\xi)) \leq \alpha(\rho(\xi)). \end{cases}$$

*(3) $g^-(t), g^+(t)$ are VHS-$\Delta$-integrable with respect to $\alpha$, and*

$$j \circ \left( (FHS) \int_a^b \tilde{f}(t)\Delta\alpha \right) = \left( (VHS) \int_a^b g^-(t)\Delta\alpha, (VHS) \int_a^b g^+(t)\Delta\alpha \right).$$

**Proof.** (1) implies (2): since $\tilde{f}(t) : [a, b]_{\mathbb{T}} \to \mathbb{R}_{\mathcal{F}}$ is FHS-$\Delta$-integrable with respect to $\alpha$ on $[a, b]_{\mathbb{T}}$, we can assume

$$\tilde{H} = \int_a^b \tilde{f}(t)\Delta\alpha.$$

Given $\varepsilon > 0$, there is a $\Delta$-gauge, $\delta$, such that for all $\delta$-fine partition of $[a, b]_{\mathbb{T}}$, we have

$$D\left( \sum \tilde{f}(\xi)(\alpha(v) - \alpha(u)), \tilde{H} \right) < \varepsilon,$$

since $\alpha$ be a function of bounded variation, then,

$$\sup_{r \in [0,1]} \max\{| \sum f_r^-(\xi)(\alpha(v) - \alpha(u)) - H_r^- |, | \sum f_r^+(\xi)(\alpha(v) - \alpha(u)) - H_r^+ |\} < \varepsilon$$

or

$$\sup_{r \in [0,1]} \max\{| \sum f_r^+(\xi)(\alpha(v) - \alpha(u)) - H_r^- |, | \sum f_r^-(\xi)(\alpha(v) - \alpha(u)) - H_r^+ |\} < \varepsilon.$$

That is,

$$D(\sum \tilde{f}(\xi)(\alpha(v) - \alpha(u)), \tilde{H}) < \varepsilon \Leftrightarrow$$
$$\sup_{r \in [0,1]} \max\{| \sum g_r^-(\xi)(\alpha(v) - \alpha(u)) - H_r^- |, | \sum g_r^+(\xi)(\alpha(v) - \alpha(u)) - H_r^+ |\}$$
$$< \varepsilon,$$

which implies

$$\begin{cases} \mid \sum g_r^-(\xi)(\alpha(v) - \alpha(u)) - H_r^- \mid < \varepsilon, \\ \mid \sum g_r^+(\xi)(\alpha(v) - \alpha(u)) - H_r^+ \mid < \varepsilon. \end{cases}$$

Thus, $g_r^-, g_r^+$ are Henstock-Stieltjes $\Delta$-integrable with respect to $\alpha$ on $[a, b]_\mathbb{T}$.

(2) implies (1): let $g_r^-, g_r^+$ be Henstock-Stieltjes $\Delta$-integrable with respect to $\alpha$ on $[a, b]_\mathbb{T}$, then there exists $H_r^-, H_r^+$ with the properties that given $\varepsilon > 0$, there is $\Delta$-gauge $\delta$ such that for any $\delta$-fine partition $P = \{[u, v]; \xi\}$ we have

$$\mid \sum g_r^-(\xi)(\alpha(v) - \alpha(u)) - H_r^- \mid < \varepsilon,$$

$$\mid \sum g_r^+(\xi)(\alpha(v) - \alpha(u)) - H_r^+ \mid < \varepsilon$$

where $r \in [0, 1]$ and $H_r^- = \int_a^b g_r^-(t)\Delta\alpha, H_r^+ = \int_a^b g_r^+(t)\Delta\alpha$.

Now we prove $\{[H_r^-, H_r^+], r \in [0, 1]\}$ represents a fuzzy number. According to Theorem 1:

(1) For $r \in [0, 1]$, if $g_r^-(t) \le g_r^+(t)$, then $H_r^- \le H_r^+$, the interval $[H_r^-, H_r^+]$ is closed;

(2) $g_r^-(t)$ and $g_r^+(t)$ are nondecreasing and nonincreasing functions on $[0, 1]$ respectively. For any $0 \le r_1 \le r_2 \le 1$,

$$\int_a^b g_{r_1}^-(t)\Delta\alpha \le \int_a^b g_{r_2}^-(t)\Delta\alpha \le \int_a^b g_{r_2}^+(t)\Delta\alpha \le \int_a^b g_{r_1}^+(t)\Delta\alpha,$$

thus $[H_{r_2}^-, H_{r_2}^+] \subset [H_{r_1}^-, H_{r_1}^+]$.

(3) Now for any sequence $\{r_n\}$ satisfies $r_n \le r_{n+1}$ and $r_n \to r \in [0, 1]$, we have

$$\bigcap_{n=1}^\infty [\tilde{f}(t)]_{r_n} = [\tilde{f}(t)]_r,$$

that implies

$$\bigcap_{n=1}^\infty [g_{r_n}^-(t), g_{r_n}^+(t)] = [g_r^-(t), g_r^+(t)],$$

and

$$\lim_{n\to\infty} g_{r_n}^-(t) = g_r^-(t), \lim_{n\to\infty} g_{r_n}^+(t) = g_r^+(t),$$

moreover,

$$g_0^-(t) \le g_{r_n}^-(t) \le g_1^-(t), g_1^+(t) \le g_{r_n}^+(t) \le g_0^+(t).$$

So, we have $g_r^-(t)$ and $g_r^+(t)$ are Henstock-Stieltjes $\Delta$-integrable with respect to $\alpha$ on $[a, b]_\mathbb{T}$ and

$$\lim_{n\to\infty} \int_a^b g_{r_n}^-(t)\Delta\alpha = \int_a^b g_r^-(t)\Delta\alpha, \lim_{n\to\infty} \int_a^b g_{r_n}^+(t)\Delta\alpha = \int_a^b g_r^+(t)\Delta\alpha.$$

Consequently,

$$\bigcap_{n=1}^{\infty} [H_{r_n}^-, H_{r_n}^+] = [H_r^-, H_r^+].$$

Now $\tilde{H} = \{[H_r^-, H_r^+], r \in [0, 1]\}$. Thus,

$$D\Big(\sum \tilde{f}(\xi)(\alpha(v) - \alpha(u)), \tilde{H}\Big) < \varepsilon,$$

that is, $\tilde{f} : [a, b]_{\mathbb{T}} \to \mathbb{R}_{\mathscr{F}}$ is FHS-$\Delta$-integrable with respect to $\alpha$ on $[a, b]_{\mathbb{T}}$.

Obviously, (3) implies (2).

(2) implies (3): since $g_r^-$ be Henstock-Stieltjes $\Delta$-integrable with respect to $\alpha$ for any $r \in [0, 1]$ uniformly, then given $\varepsilon > 0$, there is $\Delta$-gauge $\delta$ such that for any $\delta$-fine partition $P = \{[u, v]; \xi\}$ we have

$$\Big| \sum g_r^-(\xi)(\alpha(v) - \alpha(u)) - H_r^- \Big| < \frac{\varepsilon}{2}.$$

This implies

$$\sup_{r \in [0,1]} \Big| \sum_{i=1}^{n} g_r^-(\xi)(\alpha(v) - \alpha(u)) - H_r^- \Big| \le \frac{\varepsilon}{2} < \varepsilon,$$

that is

$$\Big\| \sum_{i=1}^{n} g^-(\xi)(\alpha(v) - \alpha(u)) - H^- \Big\|_X < \varepsilon$$

where $\|\cdot\|_X$ represents the norm in $\overline{C}[0, 1]$.

Since $g_0^-(t) \le g_{r_n}^-(t) \le g_1^-(t)$ and $g^-(t) \in \overline{C}[0, 1]$, by the dominated convergence theorem we infer that

$$\lim_{n \to \infty} \int_a^b g_{r_n}^-(t)\Delta\alpha = \int_a^b g_r^-(t)\Delta\alpha$$

for any nondecreasing sequence of number $r_n \to r$, and

$$\lim_{n \to \infty} \int_a^b g_{r_n}^-(t)\Delta\alpha = \int_a^b g_{r^+}^-(t)\Delta\alpha$$

for any nonincreasing sequence of number $r_n \to r$. Especially, $H_r^-$ is right continuous at 0.

This implies

$$(VHS) \int_a^b g^-(t)\Delta\alpha = H^- \in \overline{C}[0, 1].$$

For the case of $g^+(t)$, the proof is similar. Consequently, $g^-(t), g^+(t)$ are VHS-$\Delta$-integrable with respect to $\alpha$, and

$$j \circ \Big((FHS) \int_a^b \tilde{f}(t)\Delta\alpha\Big) = \Big((VHS) \int_a^b g^-(t)\Delta\alpha, (VHS) \int_a^b g^+(t)\Delta\alpha\Big).$$

The proof is complete.

**Example 3** *Let* $\mathbb{T} = \left[\frac{1}{2^{i+1}}, \frac{1}{2^i}\right], i = 1, 3, 5$. *Define a fuzzy-number-valued function* $\tilde{f}(t)(s) = t\tilde{u}(s)$, *where*

$$
\tilde{u}(s) = \begin{cases} s, & 0 \leq s \leq 1, \\ 2 - s, & 1 < s \leq 2, \\ 0, & \text{otherwise} \end{cases}
$$

*and* $\alpha(t) = kt$ *for* $t \in [0,1]_{\mathbb{T}} = [0,1] \cap \mathbb{T}, k \in \mathbb{R}$. *Obviously, the r-level sets could be calculated as follows:*

$$
[\tilde{f}(t)(s)]_r = [tr, t(2-r)].
$$

*Case 1. If* $k > 0$, *then* $\alpha(t) = kt$ *and*

$$
g_r^-(t) = tr, \quad g_r^+(t) = t(2-r).
$$

*By Definition 10,* $g^-(t) = tr \in \overline{C}[0,1], g^+(t) = t(2-r) \in \overline{C}[0,1]$. *Thus*

$$
\int_0^1 g^-(t)\Delta\alpha = \int_0^1 tr\Delta kt = \frac{819}{8192}kr,
$$

$$
\int_0^1 g^+(t)\Delta\alpha = \int_0^1 t(2-r)\Delta kt = \frac{819}{8192}k(2-r).
$$

*According to Theorem 12(3), we have*

$$
j \circ \left((FHS) \int_0^1 \tilde{f}(t)(s)\Delta\alpha\right) = \left(\frac{819}{8192}kr, \frac{819}{8192}k(2-r)\right).
$$

*Therefore,* $\tilde{f}(t)(s)$ *is FHS-$\Delta$-integral with respect to* $\alpha(t) = kt$.
*Case 2. If* $k < 0$, *then* $\alpha(t) = kt$ *and*

$$
g_r^-(t) = t(2-r), \quad g_r^+(t) = tr,
$$

$$
H_r^- = \int_0^1 t(2-r)\Delta(kt) = \frac{819}{8192}k(2-r), H_r^+ = \int_0^1 tr\Delta(kt) = \frac{819}{8192}kr.
$$

*Thus, we have*

$$
\left| \sum g_r^-(\xi)(\alpha(v) - \alpha(u)) - H_r^- \right|
$$
$$
= \left| \sum \xi(2-r)k(v-u) - \frac{819}{8192}k(2-r) \right|
$$
$$
< \varepsilon
$$

*and*

$$
\left| \sum g_r^+(\xi)(\alpha(v) - \alpha(u)) - H_r^+ \right|
$$
$$
= \left| \sum \xi rk(v-u) - \frac{819}{8192}kr \right|
$$
$$
< \varepsilon.
$$

*This means,*

$$D\left(\sum \tilde{f}(\xi)(\alpha(\nu) - \alpha(u)), \tilde{H}\right) < \varepsilon.$$

*Therefore, $\tilde{f}(t)(s)$ is FHS-$\Delta$-integral with respect to $\alpha(t) = kt$.*

## Conclusions

The aim of this paper is to extend the theory of the fuzzy Henstock-Stieltjes integral in a general sense. We first investigate the fuzzy Henstock-Stietljes $\Delta$-integral on time scales. We also obtain some properties and several necessary and sufficient conditions for fuzzy Henstock-Stieltjes $\Delta$-integrability. In addition, we also give a characterization theorem of fuzzy Henstock-Stietljes $\Delta$-integrable functions by using the isometric isomorphic embedding operator. In future research, we shall discuss some applications of the fuzzy Henstock-Stietljes $\Delta$-integral, such as fuzzy Henstock-Stietljes $\Delta$-integral on unbounded time scales and initial value problems for a class of generalized fuzzy dynamic equations on time scales.

## Author Contributions

**Formal analysis:** Juan Li, Yubing Li, Yabin Shao.

**Methodology:** Juan Li.

**Supervision:** Yabin Shao.

**Writing – original draft:** Juan Li, Yubing Li.

**Writing – review & editing:** Yabin Shao.

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
