## [Decision Letter · Decision Letter 0]

25 Apr 2024

PONE-D-24-12527On Fuzzy Henstock-Stieltjes Integral on Time Scales with respect to Bounded Variation FunctionPLOS ONE

Dear Dr. Shao,

Thank you for submitting your manuscript to PLOS ONE. After careful consideration, we feel that it has merit but does not fully meet PLOS ONE’s publication criteria as it currently stands. Therefore, we invite you to submit a revised version of the manuscript that addresses the points raised during the review process.

**ACADEMIC EDITOR: **

In this paper, the authors considered investigating the basic theory of the Henstock-Stieltjes fuzzy integral with respect to a bounded variation function on the time scale. The considered problem is interesting and the results are proven by the standard method, and the paper can be considered for publication after corrections:

1) The state of the art must be improved.

2) An extensive review of the text of the paper must be carried out. Typing errors must be eliminated, mathematical notations must be presented appropriately according to the Journal template;

3) Preliminaries about time scales should be provided in the second section;

4) Some applications of the main results must be presented;

5) The authors included many preliminaries. It would be beneficial if they provided a brief explanation of their relevance to the results;

6) Quote Definition 3.1 and others;

7) There are not enough smooth connections between results. It is recommended that authors provide a few introductory lines before stating and proving each new result;

8) If possible, could the authors include any corollaries and observations for = and T=Z?

9) The results are applicable to general time scales and therefore may be relevant to both continuous and discrete time scales. It would be beneficial to compare these results with those existing on real, discrete time scales, if possible;

10) The conclusion section is very brief. Should provide a more detailed explanation.

We look forward to receiving your revised manuscript.

Kind regards,

Angelo Marcelo Tusset

Academic Editor

PLOS ONE

Additional Editor Comments:

In this paper, the authors considered investigating the basic theory of the Henstock-Stieltjes fuzzy integral with respect to a bounded variation function on the time scale. The considered problem is interesting and the results are proven by the standard method, and the paper can be considered for publication after corrections:

1) The state of the art must be improved.

2) An extensive review of the text of the paper must be carried out. Typing errors must be eliminated, mathematical notations must be presented appropriately according to the Journal template;

3) Preliminaries about time scales should be provided in the second section;

4) Some applications of the main results must be presented;

5) The authors included many preliminaries. It would be beneficial if they provided a brief explanation of their relevance to the results;

6) Quote Definition 3.1 and others;

7) There are not enough smooth connections between results. It is recommended that authors provide a few introductory lines before stating and proving each new result;

8) If possible, could the authors include any corollaries and observations for = and T=Z?

9) The results are applicable to general time scales and therefore may be relevant to both continuous and discrete time scales. It would be beneficial to compare these results with those existing on real, discrete time scales, if possible;

10) The conclusion section is very brief. Should provide a more detailed explanation.

Reviewers' comments:

Reviewer's Responses to Questions

**Comments to the Author**

1. Is the manuscript technically sound, and do the data support the conclusions?

Reviewer #1: Yes

Reviewer #2: Yes

2. Has the statistical analysis been performed appropriately and rigorously? 

Reviewer #1: No

Reviewer #2: N/A

3. Have the authors made all data underlying the findings in their manuscript fully available?

Reviewer #1: No

Reviewer #2: Yes

4. Is the manuscript presented in an intelligible fashion and written in standard English?

Reviewer #1: Yes

Reviewer #2: Yes

5. Review Comments to the Author

Reviewer #1: In this paper, the authors considered investigate the basic theory of fuzzy Henstock-Stieltjes Δ-integral with respect to a bounded variation function on time scale. A series of mian results about fuzzy Henstock-Stieltjes Δ-integrable function were presented. Then some examples were also given to show the effectiveness of main results. The main reuslts of this paper are interesting and their proofs seem to be correct. Some speical comments are given as follows.

1) The research background of this paper should be improved.

2) Preliminaries on time scales should be provided in the second section.

3) Some applications of the main results should be given.

Reviewer #2: The considered problem is interesting, and the results are proven with the standard method. However, the representation of the paper is not satisfactory. It requires some revision. I have some suggestions and comments. Please see the attached file.

6. PLOS authors have the option to publish the peer review history of their article (what does this mean?). If published, this will include your full peer review and any attached files.

Reviewer #1: No

Reviewer #2: No

---

## [Decision Letter · Decision Letter 1]

5 Aug 2024

On Fuzzy Henstock-Stieltjes Integral on Time Scales with Respect to Bounded Variation Function

PONE-D-24-12527R1

Dear Dr. Yabin Shao,

We’re pleased to inform you that your manuscript has been judged scientifically suitable for publication and will be formally accepted for publication once it meets all outstanding technical requirements.

Kind regards,

Angelo Marcelo Tusset

Academic Editor

PLOS ONE

Additional Editor Comments (optional):

The authors presented a fully revised version, meeting all the requested corrections and the criteria required for publication of this Journal.

After these considerations, I consider the paper accepted in its current form.

Reviewers' comments:

Reviewer's Responses to Questions

**Comments to the Author**

1. If the authors have adequately addressed your comments raised in a previous round of review and you feel that this manuscript is now acceptable for publication, you may indicate that here to bypass the “Comments to the Author” section, enter your conflict of interest statement in the “Confidential to Editor” section, and submit your "Accept" recommendation.

Reviewer #1: All comments have been addressed

Reviewer #3: All comments have been addressed

2. Is the manuscript technically sound, and do the data support the conclusions?

Reviewer #1: Yes

Reviewer #3: Yes

3. Has the statistical analysis been performed appropriately and rigorously? 

Reviewer #1: Yes

Reviewer #3: Yes

4. Have the authors made all data underlying the findings in their manuscript fully available?

Reviewer #1: Yes

Reviewer #3: Yes

5. Is the manuscript presented in an intelligible fashion and written in standard English?

Reviewer #1: Yes

Reviewer #3: Yes

6. Review Comments to the Author

Reviewer #1: The authors have answered my concerned questions in the revised version. Therefore, I recommend it accepted for Plos one.

Reviewer #3: The author has made the changes requested by the reviewers. The article is good enough to be published.

7. PLOS authors have the option to publish the peer review history of their article (what does this mean?). If published, this will include your full peer review and any attached files.

Reviewer #1: No

Reviewer #3: No

---

## [Editor Report · Acceptance letter]

7 Aug 2024

PONE-D-24-12527R1 

PLOS ONE

Dear Dr. Shao, 

I'm pleased to inform you that your manuscript has been deemed suitable for publication in PLOS ONE. Congratulations! Your manuscript is now being handed over to our production team.

Kind regards, 

on behalf of

Professor Angelo Marcelo Tusset 

Academic Editor

PLOS ONE